# Investigation of the Genetic Diversity of Dagestan Mountain Cattle Using STR-Markers



**Valeria V. Volkova** [1,*], **Alexandra S. Abdelmanova** [1], **Tatiana E. Deniskova** [1], **Olga S. Romanenkova** [1], **Abdusalam A. Khozhokov** [2], **Alimsoltan A. Ozdemirov** [2], **Alexander A. Sermyagin** [1] **and Natalia A. Zinovieva** [1]

1    L.K. Ernst Federal Research Center for Animal Husbandry, Dubrovitsy 60, Podolsk Municipal District, 142132 Moscow, Moscow Region, Russia; preevetic@mail.ru (A.S.A.); horarka@yandex.ru (T.E.D.); ksilosa@gmail.com (O.S.R.); alex_sermyagin85@mail.ru (A.A.S.); n_zinovieva@mail.ru (N.A.Z.)

2    Federal State Budgetary Scientific Institution "Federal Agrarian Scientific Center of the Republic of Dagestan", Akushinsky Ave., Scientific Town, 367014 Makhachkala, Republic of Dagestan, Russia; a.mar2013@mail.ru (A.A.K.); alim72@mail.ru (A.A.O.)

*    Correspondence: moonlit_elf@mail.ru

**Abstract:** The Dagestan Mountain (DM) cattle breed was established to provide cheese and milk products to mountain dwellers in the specific conditions of the Republic of Dagestan in Southern Russia. Only 650 head of DM cattle were registered in 2020, and their "breed purity" is questionable. We aimed to assess the genetic diversity and population structure of modern DM cattle using short tandem repeat (STR) markers. The sample included 150 animals collected from private owners in Dagestan during a scientific expedition ($n = 32$) and provided by a gene pool farm ($n = 118$). An additional 166 samples from other cattle breeds distributed in the breeding zone of DM cattle were used as the comparison groups. The genotypes for the 11 STR loci recommended by ISAG were obtained using a genetic analyzer. We identified a high level of genetic diversity as revealed by allelic richness calculations (Ar = 6.82 vs. 4.38–5.82 in other cattle breeds) and observed heterozygosity indices (uHe = 0.76 vs. 0.65–0.72). Based on the STRUCTURE results, animals with low levels of admixture with other breeds were found within the DM cattle, which can be considered as candidates for use in germplasm conservation programs.

**Keywords:** cattle; local breeds; STR-markers; population structure





## 1. Introduction

Dagestan is located in the southernmost part of the Russian Federation on the northeastern slopes of the Greater Caucasus and Caspian lowlands. The territory of Dagestan comprises several landscape zones, including plains, foothills, mountains, mountain valleys, and highlands. The mountainous zone occupies almost half of the territory (48%) [1]. A study of the labor activity of the ethnic population of Dagestan showed that the source of the highlanders' livelihood was manual labor, which was complicated by unfavorable natural conditions, such as low atmospheric pressure, rarefied air, lack of oxygen, sudden temperature changes (day–night), and difficult terrain [2]. Simultaneously, large areas of mountain pastures, cheap pasture fodder, and a long pasture season favor cattle breeding. The inhabitants of the plain and foothill zones prefer cattle breeding, whereas mountain dwellers raise both sheep and cattle. Livestock management systems differ depending on the landscape zone. Combined pasture and stall-cattle keeping are traditional management systems in the Dagestan flatlands. From spring to late autumn, the cattle are kept on rural pastures with no feed supplements. Residents of some communities move cattle from rural to mountain pastures from spring to autumn, when raw milk is processed into butter, cheese, and cottage cheese. A transhumance system is used in the mountains [2].

The Republic of Dagestan has a long history of livestock husbandry. Great Caucasian and Lesser Caucasian cattle were raised in the highlands and flatlands until the 1930s;

currently, these cattle are reared in several regions. The Dagestan Mountain (DM) breed was obtained by crossings of local and brown Swiss cattle with further improvement by the Kostroma and Lebedinsky breeds [3] (Figure 1).

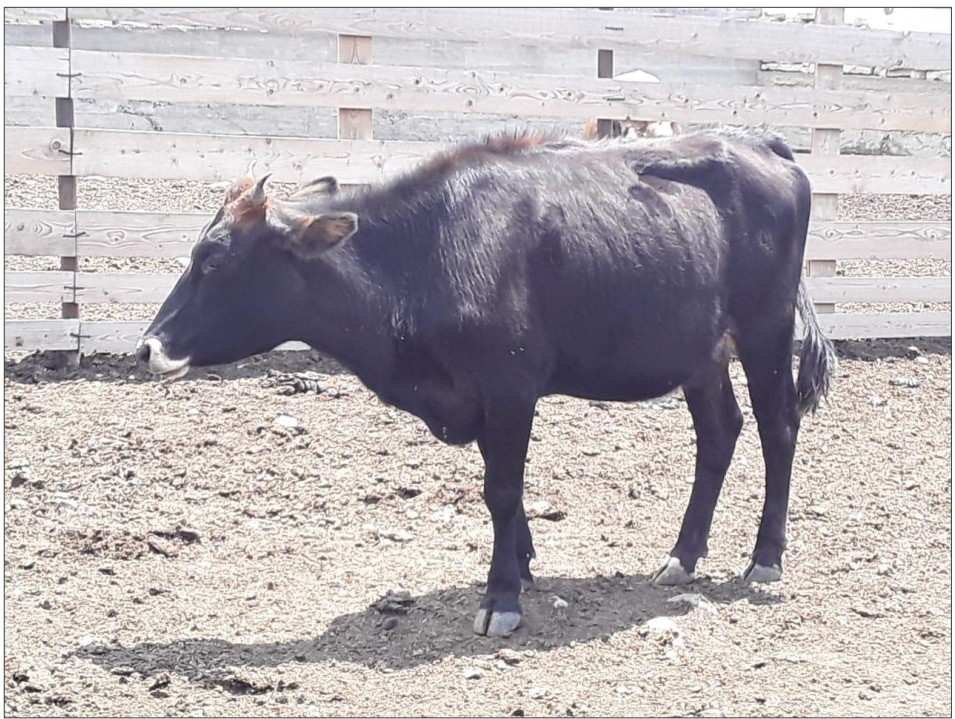

**Figure 1.** Typical Dagestan Mountain cow (Republic of Dagestan). Photo provided by Alexander A. Sermyagin.

DM cattle represent a valuable gene pool, and are well adapted to breeding in extremal highlands in the climate of southern Russia. Owing to their comparably low body weight, DM cattle effortlessly move around the highlands and consume vegetation in places that are inaccessible to other cattle breeds. In addition to adapting to environmental conditions, the most noticeable features of mountain cattle are their obedience and their ability to herd without human supervision. They are released from the byre in the morning to find their way through the outskirts of villages and mountain slopes as they forage for food. In the evening, they return to the byres [4]. These features facilitate the obtention of milk and meat from these animals without spending any resources on care and feeding. These valuable qualities allowed highlanders to survive harsh conditions for many centuries.

The human population has been growing steadily since the 20th century [5]. Extensive livestock farming can no longer meet the demand for animal proteins. Highland cattle cannot compete with highly productive commercial breeds owing to their low milk yield and fattening ability. Therefore, the breeding of these animals has not received proper attention in the Republic of Dagestan. Poor fodder resources and insufficient selection are the main reasons for the low productivity of DM cattle [6]. The intensification of cattle breeding is unsuccessful because of the specific natural geographic conditions in highland areas. The breeding of specialized commercial cattle in mountainous terrain leads to a considerable decrease in milk yield and early culling for various reasons [7]. In this regard, conservation of the DM cattle gene pool is an important priority for livestock husbandry in mountainous terrains.

Only 650 head of DM cattle were registered in 2020 and kept at a single gene pool farm in Dagestan. Most cattle lack complete pedigree records; therefore, their "breed purity" is questionable. We aimed to characterize the current gene pool, evaluate genetic diversity, and analyze the population structure of DM cattle in comparison with other cattle breeds reared in Russia based on microsatellite analysis.

## 2. Materials and Methods

### 2.1. Sample Collection

Tissue samples of DM cattle (*n* = 32) were collected from private owners in the mountain villages of Dagestan during a scientific expedition in 2019. In 2021, the gene pool farm involved in the conservation of DM cattle provided 118 additional samples (Supplementary Table S1 and Supplementary Figure S1). The final sample consisted of 150 specimens. The short tandem repeat (STR) genotypes of five other cattle breeds (*n* = 166) bred in this breeding zone, which could have contributed to the gene pool of DM cattle, were obtained from the Bioresource Collection of Farm Animals of the L.K. Ernst Federal Science Center for Animal Husbandry, supported by the Russian Ministry of Science and Higher Education [8] (Table 1).

**Table 1.** Sampling information for studied cattle breeds.

| Breed | Code | *n* | Breeding Region |
|---|---|---|---|
| Dagestan Mountain | DM | 150 | The Republic of Dagestan |
| Brown Swiss, Caucasian branch | BS_D | 13 | The Republic of Dagestan |
| Brown Swiss, Germany | BS_G | 27 | Germany |
| Red Steppe | RS | 26 | The Republic of Dagestan |
| Simmental | SIM | 50 | Oryol region |
| Holstein [1] | HOL | 50 | Holstein Association USA, Inc. |

[1] STR-genotypes of Holstein cattle breed were obtained from the Holstein Association USA, Inc. (Brattleboro, VT, USA).

### 2.2. DNA Extraction and STR-Genotyping

The genomic DNA was extracted using the DNA-Extran-2 and S-Sorb Kits (SyntolTM, Moscow, Russia) according to the manufacturer's instructions.

In this study, we used eleven microsatellite loci, which were recommended by the International Society for Animal Genetics (ISAG) [9] and included BM1818, BM2113, BM1824, [10], ETH10 [11], ETH225 [12], INRA023 [13], SPS115 [14] and TGLA53, TGLA122, TGLA126, TGLA227 [15]. All microsatellite loci were autosomal. The 5′ fluorescently labeled primer sequences and chromosomal localization of each microsatellite locus are provided in Supplementary Table S2.

PCR amplification was performed according to standard protocol [16]. The total reaction volume of 9 μL comprised 1 μL 10× PCR-buffer, 1 μL dNTP (1 mmol/L), 0.5 μL MgCl2, 0.34 μL of primer master mix, 0.1 μL of Smart Taq DNA polymerase (Dialat ltd, Moscow, Russia), and 1 μL of DNA template. The PCR amplification included initial denaturation (95 °C for 10 min), 40 cycles (95 °C for 30 s, 60 °C for 40 s, and 72 °C for 1 min), and a final extension (72 °C for 10 min).

Microsatellite analysis was performed using ABI 3130xl Genetic Analyzer (Applied Biosystems, Waltham, MA, USA). Raw allele sizes were determined using the GeneMapper v.4.0 software (Applied Biosystems, Waltham, MA, USA).

### 2.3. Data Analysis

Allelic and genetic diversity parameters were calculated using GenAIEx 6.503 software [17] and the R package diversity [18].

To study the genetic differentiation between DM and other cattle populations, we calculated pairwise $F_{ST}$ values [19] and Jost's index of population differentiation (Jost's D) [20] using the R Project for Statistical Computing software environment [21].

The Neighbor-net graph, which is based on Jost's index of population differentiation, was constructed using SplitsTree 4.14.5 software [22].

Principal component analysis (PCA) was performed using the R package adegenet [23] and visualized using the R package ggplot2 [24]. The map with sampling sites was created using R package maps [25]. Data files were prepared using R 3.5.0 [21].

The genetic structure of the populations was estimated in Structure 2.3.4 software [26] using the following parameters: burn-in period—10,000; number of Markov chain Monte Carlo simulations (MCMC)—100,000 for each run. Ten iterations were performed for each K. CLUMPAK software [27,28] was used to visualize and to determine the most probable number of clusters in the studied sample based on the ΔK values according to the method proposed by Evanno et al. [29]. To analyze the population structure, we determined the average similarity scores produced in CLUMPAK for several independent runs with the same K-value.

## 3. Results

### 3.1. Genetic Variation among and within Breeds

The statistics of the analyzed loci are summarized in Table 2. The total number of alleles per locus ranged from 26 (BM1824) to 61 (TGLA53) alleles.

**Table 2.** Allelic variability in studied breeds.

| Locus | Population | | | | | | In Total [1] |
|---|---|---|---|---|---|---|---|
| | DM | BS_D | BS_G | RS | HOL | SIM | |
| TGLA227 | 12/1 | 7 | 6 | 9 | 7/1 | 8 | 49/2 |
| BM2113 | 11/4 | 6 | 7 | 6 | 5 | 5 | 40/4 |
| TGLA53 | 17/2 | 9 | 9 | 8 | 7 | 11 | 61/2 |
| ETH10 | 8/1 | 5/1 | 4 | 5 | 6 | 5 | 33/2 |
| SPS115 | 9/2 | 4 | 5 | 6 | 4 | 6 | 34/2 |
| TGLA122 | 17/6 | 6 | 8/2 | 7/1 | 7/1 | 9 | 54/10 |
| INRA23 | 11/1 | 7 | 5 | 8 | 4 | 8/1 | 43/2 |
| TGLA126 | 8/3 | 4 | 3 | 4 | 4 | 6/1 | 29/4 |
| BM1818 | 8 | 5 | 6 | 3 | 4 | 6 | 32/0 |
| ETH225 | 11/4 | 7/1 | 5 | 7 | 5 | 6 | 41/5 |
| BM1824 | 6/1 | 4 | 4 | 4 | 3 | 5 | 26/1 |
| Total | 118/25 | 64/2 | 62/2 | 67/1 | 56/2 | 75/2 | 442/34 |

[1] Number of alleles/number of private alleles.

In total, 442 alleles across 11 loci and 34 private alleles across 10 loci were identified in the studied populations. No private alleles were found at the BM1818 locus. The DM population was characterized by the highest number of alleles (118 alleles, including 25 private alleles).

Table 3 presents the main statistical indicators used to estimate the current allele pool and the level of genetic diversity of the studied cattle populations.

**Table 3.** Genetic diversity parameters estimated for 11 microsatellite markers.

| Population | n [1] | Ar [2] (M ± SE) [7] | Ho [3] (M ± SE) | uHe [4] (M ± SE) | uFis [5] [CI] [6] |
|---|---|---|---|---|---|
| DM | 150 | 6.827 ± 0.654 | 0.723 ± 0.032 | 0.764 ± 0.031 | 0.052 [0.005; 0.099] |
| BS_D | 13 | 5.818 ± 0.483 | 0.692 ± 0.069 | 0.713 ± 0.055 | 0.023 [−0.093; 0.139] |
| BS_G | 27 | 5.146 ± 0.444 | 0.741 ± 0.045 | 0.701 ± 0.039 | −0.058 [−0.111; −0.005] |
| RS | 26 | 5.278 ± 0.450 | 0.692 ± 0.054 | 0.718 ± 0.026 | 0.046 [−0.059; 0.151] |
| HOL | 50 | 4.376 ± 0.396 | 0.669 ± 0.044 | 0.645 ± 0.039 | −0.038 [−0.110; 0.034] |
| SIM | 50 | 5.307 ± 0.373 | 0.729 ± 0.033 | 0.682 ± 0.031 | −0.073 [−0.115; −0.031] |

[1] *n*, number of individuals; [2] Ar, rarefied allele richness; [3] Ho, observed heterozygosity; [4] uHe, unbiased expected heterozygosity; [5] uFis, unbiased inbreeding coefficient; [6] CI, variation range of coefficient of uFis at a confidence interval of 95%; [7] M ± SE, mean value and standard error.

We observed the highest level of genetic diversity in DM cattle compared to other breeds, as revealed by allelic richness (Ar = 6.827 ± 0.654 vs. 4.376–5.818) and unbiased expected heterozygosity (uHe = 0.764 ± 0.031 vs. 0.645–0.718). The breeds raised in the Caucasus showed a higher level of genetic diversity (Ar values varied from 5.278 ± 0.45 to 6.827 ± 0.654) compared to transboundary breeds (Ar = 4.376 ± 0.396 to 5.307 ± 0.373),

which can be the consequence of lower selection pressure. We detected a larger heterozygote deficiency in DM cattle (uFis = 0.052), which can be caused by the low population size. The SIM and BS_G groups revealed an excess of heterozygotes (uFis = −0.073 and uFis = −0.058, respectively), while the heterozygote number of other cattle breeds did not show a significant deviation from the Hardy–Weinberg equilibrium.

### 3.2. Genetic Differentiation

Table 4 shows Jost's D genetic distances and $F_{ST}$ values, which characterize the degree of differentiation between the studied cattle populations, calculated for each pair of groups.

**Table 4.** Genetic differentiation between the DM and other studied cattle breeds.

| Population | DM | BS_D | BS_G | RS | HOL | SIM |
|---|---|---|---|---|---|---|
| DM | 0 | 0.023 [1] | 0.053 [1] | 0.036 [1] | 0.113 [1] | 0.058 [1] |
| BS_D | 0.049 [2] | 0 | 0.047 [1] | 0.049 [1] | 0.135 [1] | 0.093 [1] |
| BS_G | 0.130 [2] | 0.054 [2] | 0 | 0.081 [1] | 0.134 [1] | 0.116 [1] |
| RS | 0.066 [2] | 0.043 [2] | 0.150 [2] | 0 | 0.105 [1] | 0.084 [1] |
| HOL | 0.271 [2] | 0.261 [2] | 0.250 [2] | 0.189 [2] | 0 | 0.138 [1] |
| SIM | 0.114 [2] | 0.195 [2] | 0.210 [2] | 0.132 [2] | 0.234 [2] | 0 |

[1] $F_{ST}$ values are presented above the diagonal; [2] Jost's D values are presented below the diagonal.

According to the classification [30], most pairwise $F_{ST}$ values between the studied cattle populations corresponded to moderate genetic differentiation (0.053–0.138). We established that the BS_G, HOL, and SIM groups had the greatest genetic distance from the other populations.

The DM cattle demonstrated moderate differentiation in all groups, with the least genetic distance from the BS_D group ($F_{ST}$ = 0.023), indicating the participation of DM cattle in the development of the Caucasian type of Brown Swiss cattle.

PCA (Figure 2a) showed that the first principal component (PC1), responsible for 5.62% of the genetic variability, clearly separated the Holstein breed from the other breeds. The second principal component (PC2) accounted for 3.83% of the genetic variability and separated the Simmental breed from the group of breeds, which were reared mainly in the Caucasus (DM, BS, and RS). The HOL and BS_G breeds formed more compact clusters, which may indicate higher selection pressure in these groups. Brown and red cattle populations, which are bred in the Caucasus, are characterized by greater genetic diversity than the transboundary breeds.

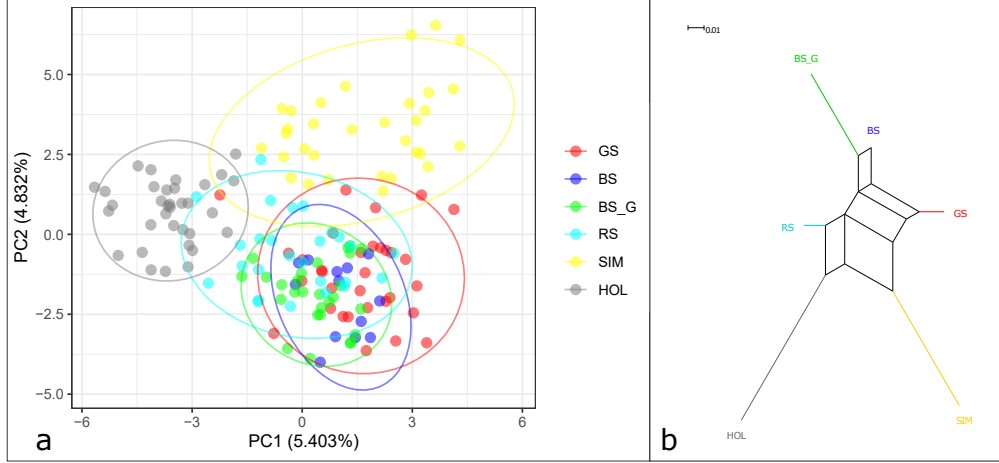

**Figure 2.** Genetic relationships between the studied cattle breeds: (**a**) Principal component analysis, X-axis: first component (PC1). Y-axis: second component (PC2); (**b**) dendrogram based on pairwise genetic distances (DJost), plotted using the Neighbor-net algorithm.

Analysis of the Neighbor-net graph (Figure 2b) showed the presence of genetic links between the DM, BS_D, and RS groups and provided evidence of the common origin of the BS_D and BS_G groups. The localization of the DM cattle at the edge of the Neighbor-net graph suggests its mixed origin. The HOL and SIM breeds form separate branches.

An analysis of the population structure (Figure 3) was performed using Structure for the number of clusters K from two to six. According to the algorithm proposed by Evanno et al. [29], the most probable number of clusters was three. At K = 2 and K = 3 (Supplementary Figure S2), we identified clusters corresponding to HOL and SIM breeds. With a further increase in K, at K = 6, BS, RS, HOL, and SIM groups formed clusters, whereas DM cattle revealed a mixed origin from several ancestral populations. We identified DM antagonists in the Dagestan population of Brown Swiss cattle, which reflects its contribution to the development of the allele pool BS_D cattle. Most of the DM animals did not show common ancestors with other studied breeds, while in a small number of animals, the admixture with BS and RS ancestors was visible. As a result of Structure analysis, DM samples were divided into two clusters. One was linked to the samples collected in the remote mountainous area, and the other was associated with the gene pool population of DM cattle. Some animals showed mixed origins.

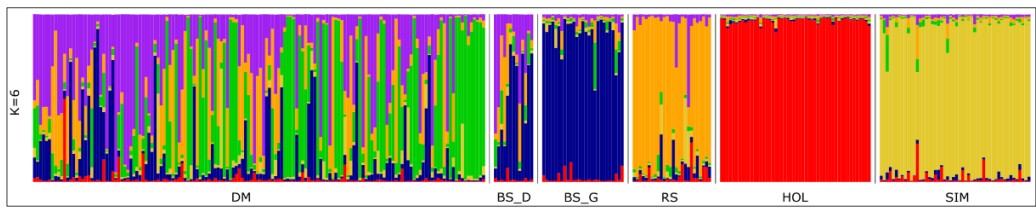

**Figure 3.** Population structure of the studied cattle breeds. DM—Dagestan Mountain cattle; BS_D—sBrown Swiss, Caucasian branch; BS_G—Brown Swiss, Germany; RS—Red Steppe; SIM—Simmental; HOL—Holstein.

## 4. Discussion

Economic significance, the high demand for dairy products and beef, and diverse environmental conditions have promoted the development of various local cattle breeds in Russia [31]. The breeds that display high productivity and satisfactory climate resilience settled widely outside the regions of their origin, while the other breeds are reared in their historical locations and are in a critical condition as a result of their small population sizes [3]. In addition, endemic breeds inhabit regions with extremely harsh environments and cannot be substituted by other local or commercial breeds without a significant increase in feeding and maintenance costs. DM cattle belong to the last group and dwell predominantly at high altitudes.

The Republic of Dagestan is a mountainous area in the Caucasus, with an abundance of natural highland and flatland pastures. Human tribes have inhabited the Caucasus since the Early Upper Paleolithic Age and most likely imported their cattle [32,33]. However, scientific data on the origin and history of DM cattle are limited and ambiguous. DM cattle belong to a branch of the Greater Caucasian group, which originated from the old dwarf cattle of Egypt [34]. The Greater Caucasian group (or Great Caucasus) included several local types, which were raised in different climatic zones within the mountainous area in the former USSR (Russia, Georgia, and Armenia) [35]. However, another study showed that local cattle, which were improved by the Brown Swiss, Kostroma, and Lebedinsky breeds, were transformed into the Lesser Caucasus cattle type [35], which corresponds to the official breed description [3].

Most local Russian cattle breeds have been investigated using STR [36,37] and single nucleotide polymorphism (SNP) markers [38–41]. However, little is known about the genetic diversity and population structure of DM cattle inhabiting the high-altitude warmer area of the Republic of Dagestan in southern Russia. Here, we present the results of the first genetic assessment of DM in cattle based on 11 microsatellite markers.

The level of genetic diversity of the DM cattle was the highest among the studied cattle populations and was compatible with the estimates obtained for other aboriginal cattle, including Indian cattle (Ho = 0.681–0.721, He = 0.702–0.751) [42], Creole (Ho = 0.719, He = 0.739) [43], and Taiwanese local cattle breeds (Ho = 0.530–0.607, He = 0.697–0.719) [44]. The DM group showed a moderate level of heterozygote deficiency. Such a pattern is frequent in native breeds [45–48], which may correspond to a lack of sires [42].

According to the classification of Hartl [30], most of the $F_{ST}$ values among the studied cattle populations exhibit a moderate level of genetic differentiation, which agrees with the patterns of genetic distances between local breeds in other countries [44,48]. Pairwise $F_{ST}$ values between Indian local cattle populations were lower ($F_{ST}$ = 0.008–0.044) [42] than those obtained in our study between DM and other Russian cattle populations, which probably reflects higher selection pressure.

In our study, we found that the DM cattle group had the lowest differentiation evaluated by $F_{ST}$ and Jost's D-values with the Brown Swiss population raised in the same breeding zone. The definite genetic closeness between these groups was demonstrated using the Neighbor-net graph (Figure 2b). In addition, shared genetic components were identified in the DM and Brown Swiss groups using structural analysis (Table 3). Several animals within the DM population («endemic groups») demonstrated the presence of a genetic background, which was absent or found in barely noticeable traces in other studied cattle groups (Figure 3). These individuals can be considered valuable national genetic resources that retain most of their ancestral genetic components. Further studies using more powerful molecular genetic instruments at the genome-wide level are necessary to determine the source of this ancestry more precisely.

## 5. Conclusions

Our study provides data on the genetic diversity, population structure, and genetic differentiation of modern DM cattle for the first time. Conservation of valuable gene pools of DM cattle is necessary for the further development of animal husbandry in the Dagestan Mountains, as well as for the breeding and selection of animals with high productivity and adaptation to harsh conditions in the mountain areas. Our studies will be continued with a larger sample size using SNP genotyping arrays and whole-genome sequencing.

**Supplementary Materials:** The following supporting information can be downloaded at: https://www.mdpi.com/article/10.3390/d14070569/s1, Figure S1: The map with sampling sites of Dagestan Mountain cattle; Figure S2: Population structure of the studied cattle breeds for K from 2 to 6; Table S1: STR-genotypes for the animals of Dagestan Mountain cattle; Table S2: The 5′ fluorescently labeled primer sequences and chromosomal localization of each microsatellite locus used in this study.

**Author Contributions:** Conceptualization, N.A.Z. and V.V.V.; methodology, V.V.V., A.S.A. and A.A.S.; software, V.V.V., A.S.A. and A.A.S.; validation, V.V.V. and O.S.R.; formal analysis, V.V.V., A.S.A., T.E.D. and A.A.S.; investigation, V.V.V., O.S.R., A.A.S., A.A.K. and A.A.O.; resources, N.A.Z.; data curation, A.S.A., V.V.V. and T.E.D.; writing—original draft preparation, V.V.V., T.E.D., A.S.A. and O.S.R.; writing—review and editing, V.V.V., A.S.A., N.A.Z., T.E.D., A.A.S. and A.A.K.; supervision, N.A.Z.; project administration, N.A.Z.; funding acquisition, N.A.Z. All authors have read and agreed to the published version of the manuscript.

**Funding:** This research was funded by the Ministry of Science and Higher Education of the Russian Federation (theme No. FGGN-2022-0002).

**Institutional Review Board Statement:** The animal study protocol was approved by the Ethics Committee of the L.K. Ernst Federal Research Center for Animal Husbandry (protocol № 1B of 21 January 22).

**Data Availability Statement:** The DNA genotypes of Dagestan Mountain cattle used in this study are presented in Supplementary Table S1. The DNA genotypes for other cattle breeds are available on request to the corresponding author.

**Acknowledgments:** To conduct this study, we used the equipment of the Center for Collective Use "Bioresources and Bioengineering of Farm Animals" of L.K. Ernst Federal Research Center for Animal Husbandry.

**Conflicts of Interest:** The authors declare no conflict of interest. The Ministry of Science and Higher Education of the Russian Federation had no role in the design of the study; in the collection, analysis, or interpretation of data; in the writing of the manuscript, or in the decision to publish the results.

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
