# Peer review of "Investigation of the Genetic Diversity of Dagestan Mountain Cattle Using STR-Markers"

_diversity, doi:10.3390/d14070569_

Round 1
Reviewer 1 Report
Abstract: No comment
Introduction: No comment
Material and methods: Section 2.3, Data Analysis, line 113, Each steps of data analysis can be mentioned by using bullet points.
Results: Is there any problem to perform whole genome sequencing. Then it will be more precise to study markers all over the genome.
Discussion: No comment
Author Response
Point 1: Material and methods: Section 2.3, Data Analysis, line 113, Each steps of data analysis can be mentioned by using bullet points.
Response 1: Dear Reviewer,
Thank you for your valuable recommendation. We will consider it in our further studies.
Point 2: Results: Is there any problem to perform whole genome sequencing. Then it will be more precise to study markers all over the genome.
Response 1: Whole genome sequencing is expensive analysis. We can perform WGS for a limited number of samples. In this regard, microsatellite analysis was planned as initial stage to study genetic structure of Dagestan Mountain cattle and to select the most typical animals to be sequenced. We added a little clarification in the Conclusions.
Thank you very much for your valuable comments, which helped us to improve our manuscript!

Reviewer 2 Report
The publication 'Investigation of the genetic diversity of Dagestan Mountain cattle using STR-markers' is legibly and clearly written. The authors clearly described the aims, presented the obtained results and conducted the discussion of the obtained results in comparison to the available literature. I did not notice any errors, therefore I recommend this manuscript for publication in "Diversity" without corrections.
Author Response
Dear Reviewer, thank you very much for your consideration!
We are greatly appreciate your positive feedback.
Reviewer 3 Report
1. Can you add one classical picture of DM bull?
2. The sample number from breeding zones is 130 or 166? Check it.
3. P3L107: The PCR cycles is 25 cycles is enough.
4. P4L141 in Table 2 Locus BM1818: Should it be 32 / 0?
5. P6L206: Does it base on STRUCTURE about “One lineage wan linked to the sample…”?
6. Check spellings: Fst in the context.
7. P7L251: Please consider it again about “more rigorous breeding work”?
Author Response
Point 1: Can you add one classical picture of DM bull?
Response 1: Dear Reviewer, Thank you very much for your valuable comments, which helped us to improve our manuscript!
We did our best to find the picture of typical DM bull. Unfortunately, we could obtain only pictures of cows.
Point 2: The sample number from breeding zones is 130 or 166? Check it.
Response 2: The sample number is 166. We corrected misspelling.
Point 3: P3L107: The PCR cycles is 25 cycles is enough.
Response 3: Thank you for the recommendation. We agree that 25 cycles are usually enough. However, while testing our PCR conditions we experimentally found that using of 40 cycles provide the best PCR results. Probably, this might be associated with the specifics of the regional synthesis of reagents.
Point 4: P4L141 in Table 2 Locus BM1818: Should it be 32 / 0?
Response 4: Yes, we corrected it. Thank you for the comment.
Point 5: P6L206: Does it base on STRUCTURE about “One lineage wan linked to the sample…”?
Response 5: Yes, it is based on STRUCTURE results. We re-phrased it as following: «As a result of Structure analysis, DM samples were divided into two clusters. One was linked to the samples collected in the remote mountainous area…»
Point 6: Check spellings: Fst in the context.
Response 6: We checked relevant spellings.
Point 7: P7L251: Please consider it again about “more rigorous breeding work”?
Response 7: We changed this phrase as « higher selection pressure».
We would like to express our sincere gratitude for the valuable comments.

Reviewer 4 Report
The justification for using microsatellite for investigation of population genetics is based on publications from the late 80’s/early 90s. Given the drastic advances in genetic technology, the authors need to discuss the limits of microsatellite markers and why they opted to choose this as their methodology.
Please provide the primer information of microsatellite loci? Are you using fluorescent primers or normal primers? Also, please mark all loci separately from autosomal or sex chromosomes?
Author Response
Point 1: The justification for using microsatellite for investigation of population genetics is based on publications from the late 80’s/early 90s. Given the drastic advances in genetic technology, the authors need to discuss the limits of microsatellite markers and why they opted to choose this as their methodology.
Response 1: Dear Reviewer, thank you very much for your valuable comments, which helped us to improve our manuscript!
We agree that microsatellites are being neglected in favor of SNP-arrays (DNA chips) and whole-genome sequencing data. However, both SNP-genotyping with using DNA chips and whole genome sequencing are expensive analyses. We can perform these analyses for a limited number of samples. In this regard, microsatellite analysis was planned as initial stage to study genetic structure of Dagestan Mountain cattle and to select the most typical animals to be sequenced. In addition, Laoun, A. et al (2020) performed comparison between SNP and microsatellite performance in different livestock species and provided evidence that «microsatellites may still be a very appropriate solution to evaluate, in a first stage, the general state of livestock at national scales». This is compatible with our present study.
Point 2: Please provide the primer information of microsatellite loci? Are you using fluorescent primers or normal primers? Also, please mark all loci separately from autosomal or sex chromosomes?
Response 2: Please find the information on the 5´ fluorescently labeled primer sequences and chromosomal localization of each microsatellite locus in Sup_Table_2. We used fluorescent primers. All loci were autosomal.
We would like to express our sincere gratitude for the valuable comments.
